# Seroprevalence of *Aspergillus*-Specific IgG Antibody among Mozambican Tuberculosis Patients

**DOI:** 10.3390/jof7080595

**Published:** 2021-07-23

**Authors:** Helmut J. F. Salzer, Isabel Massango, Nilesh Bhatt, Emelva Machonisse, Maja Reimann, Sven Heldt, Christoph Lange, Michael Hoelscher, Celso Khosa, Andrea Rachow

**Affiliations:** 1Department of Pulmonary Medicine, Kepler University Hospital, 4021 Linz, Austria; salzer.helmut@gmail.com (H.J.F.S.); sven.heldt@kepleruniklinikum.at (S.H.); 2Instituto Nacional de Saúde (INS), Marracuene 3934, Mozambique; isabel.timana@gmail.com (I.M.); nbhatt.mz@gmail.com (N.B.); emelvamanhica@gmail.com (E.M.); 3Division of Clinical Infectious Diseases, Research Center Borstel, 23845 Borstel, Germany; mreimann@fz-borstel.de (M.R.); clange@fz-borstel.de (C.L.); 4German Center for Infection Research (DZIF), Partner Site Hamburg-Lübeck-Borstel-Riems, 23845 Borstel, Germany; 5Respiratory Medicine & International Health, University of Lübeck, 23563 Luebeck, Germany; 6Baylor College of Medicine and Texas Children’s Hospital, Houston, TX 77030, USA; 7Division of Infectious Diseases and Tropical Medicine, University Hospital, LMU Munich, 80802 Munich, Germany; hoelscher@lrz.uni-muenchen.de; 8German Center for Infection Research (DZIF), Partner Site Munich, 80802 Munich, Germany; 9Center for International Health-CIH LMU, 80802 Munich, Germany

**Keywords:** *Aspergillus* IgG antibody, seroprevalence, chronic pulmonary aspergillosis (CPA), tuberculosis (TB), human immunodeficiency virus (HIV), Mozambique

## Abstract

Background: Chronic pulmonary aspergillosis (CPA) is a life-threatening sequel in patients with pulmonary tuberculosis (PTB). *Aspergillus*-specific IgG antibody is a useful diagnostic biomarker supporting CPA diagnosis, especially in countries with limited health recourses. Methods: We conducted a prospective pilot study to assess the seroprevalence of *Aspergillus*-specific IgG antibodies among 61 Mozambican tuberculosis patients before, during, and after the end of TB treatment. *Aspergillus*-specific IgG antibody levels were measured using the ImmunoCAP^®^. Results: In this study, 3 out of 21 HIV-negative PTB patients had a positive *Aspergillus*-specific IgG antibody level before, during, and after the end of TB treatment. Antibody levels were 41.1, 45.5, and 174 mg/L at end of treatment (EOT), respectively. Additionally, two HIV-negative PTB patients with negative *Aspergillus*-specific IgG antibody levels at baseline became seropositive at EOT (41.9 and 158 mg/L, respectively). Interestingly, none of the HIV-positive PTB patients (40/61) had a positive *Aspergillus*-specific IgG antibody level at any time, neither at baseline nor at EOT. Probable CPA was diagnosed in one HIV-negative patient (5%; 1/20). Conclusion: Seroprevalence of *Aspergillus*-specific IgG antibody may differ between HIV-negative and HIV-positive Mozambican PTB patients. Future studies evaluating post-tuberculosis lung disease should integrate CPA as a life-threatening sequel to PTB.

## 1. Introduction

Chronic pulmonary aspergillosis (CPA) is a significant cause of morbidity and mortality in patients with respiratory disorders. The 5-year case fatality rate is around 45%, depending on the underlying respiratory disorder and associated comorbidities [1,2]. Pulmonary tuberculosis (PTB) is the most important risk factor to develop CPA. The 5-year prevalence of CPA as a sequel to PTB is estimated with around 1.2 million patients worldwide [3]. A considerable number of these patients are coming from Africa since 25% of all new TB infections worldwide were reported from the African continent in 2019 [4]. Global efforts including the “End TB Strategy” and the political declaration of the UN high-level meeting on TB strongly focus on the reduction of the TB incidence rate and the number of TB-related death, while life-threatening long-term complications due to TB including CPA have been neglected so far [5]. There is limited evidence on CPA/PTB co-infection, but a recent systematic review and meta-analysis reported *Aspergillus* coinfection in 15% of patients with PTB in Asian and African countries [6].

The diagnosis of CPA remains challenging and is based on four criteria, including a CPA-typical radiological pattern, any direct or indirect mycological evidence, a chronic course of the disease, and the exclusion of an alternative diagnosis [7,8,9]. Radiological patterns and clinical presentation often overlap between CPA and active or post-PTB. Proof of mycological evidence can make the difference, but fungal culture or nucleic acid amplification is usually not available in resource-constrained settings. Implementation of an enzyme immunoassay for the detection of *Aspergillus*-specific IgG antibodies is a promising approach to provide evidence for mycological infection [10,11]. There are several commercially available *Aspergillus*-specific IgG antibody tests showing high sensitivity and specificity of up to 93% and 99%, respectively [12,13]. Whether these immunological tests show comparable performance in PTB patients positive for human immunodeficiency virus (HIV) remains unclear, while HIV is a leading determinate for TB. Therefore, the relevance of HIV in middle- and high-incidence countries for TB must be considered to allow correct interpretation of serological test results and to allow correct CPA diagnosis.

The aim of this prospective pilot study was to analyze and compare the seroprevalence of *Aspergillus*-specific IgG antibodies among HIV-positive and -negative Mozambicans with PTB before, during, and after the end of TB treatment. Furthermore, the frequency of probable CPA as a sequel to PTB after the end of treatment (EOT) should be assessed.

## 2. Material and Methods

### 2.1. Study Design and Setting

A prospective cohort study was performed at the Instituto Nacional de Saúde (INS) TB Research Clinic in Maputo, Mozambique. The study was conducted as a pilot study to collect information for a large-scale TB sequel study [14]. Therefore, no sample size calculation was performed prior to the start of the investigation.

### 2.2. Population and Inclusion Criteria

Patients with newly diagnosed PTB were consecutively enrolled during the study period and prospectively followed for at least 52 weeks. Active PTB was defined by a positive Xpert MTB/RIF test result from sputum. All patients were tested for HIV infection. PTB patients aged 18 years having at least a serum sample at baseline (=time of diagnosis ± 8 weeks) and, after the end of treatment (EOT), were included for analyses in this study.

### 2.3. Study Assay and X-ray

Measurements of *Aspergillus*-specific IgG antibody levels were performed using the automated ImmunoCAP^®^ (Phadia 100, Thermo Fisher Scientific, Uppsala, Sweden). An *Aspergillus*-specific IgG antibody with cutoff ≥ 40 mg/L was considered positive, as recommended by manufacturer and as used by previous published studies. X-rays were performed at baseline and at EOT after 6 months of TB treatment.

### 2.4. Probable CPA

PTB patients with a CPA-typical radiological pattern based on current recommendations including one or more cavities with or without a fungal ball present or nodules after EOT and a positive *Aspergillus*-specific IgG antibody level were defined as probable CPA [7,15].

### 2.5. Statistical Analysis

Data Analysis was performed using R v. 4.0.2. For the general data analysis, fixed-effects models were used to remove the temporal effect. In the figures, the Wilcoxon rank-sum test was used to graphically represent statistically significant differences. The calculation of the slope changes is also subject to the linear mixed-effects model with restricted maximum likelihood (REML) method.

### 2.6. Ethics

All research procedures, including the consenting process, were approved by Comité Nacional de Bioética para Saúde (CNBS), Mozambique, with the reference 318/CNBS/18 and the Ethics Commission of the Medical Faculty at Ludwig Maximilian University, Germany.

## 3. Results

Of the 69 patients with PTB that were enrolled from 14 June 2014 to 28 May 2015 into the main study, 8 PTB patients were excluded from this analysis because of absent serum samples at baseline and/or after EOT, remaining with 61 PTB patients. Baseline characteristics are shown in Table 1. Cavitary PTB was present on chest X-ray in 50% (20/40) of patients at baseline and in 33% (14/43) of patients after 6 months of tuberculosis treatment, respectively.

The baseline median *Aspergillus*-specific IgG antibody level was 7.37 mg/L (IQR 2.0–158.0) for all PTB patients (Figure 1). HIV-positive individuals had a lower baseline median *Aspergillus*-specific IgG antibody level of 6.72 (IQR 2.36–17.4), while HIV-negative individuals had a higher baseline median *Aspergillus*-specific IgG antibody level of 11.65 (IQR 2.0–158.0) (Figure 2).

After EOT the median *Aspergillus*-specific IgG antibody level increased to 9.08 mg/L (IQR 2.05–174.0) for all patients. Again HIV-positive individuals had a lower median *Aspergillus*-specific IgG antibody level of 7.55 mg/L (IQR 2.05–20.6), compared to HIV-negative individuals, who had a higher median *Aspergillus*-specific IgG antibody level of 16.1 mg/L (IQR of 3.09–174.0) at week 26 and 52, respectively (Figure 3).

None of the HIV-positive PTB patients had a positive *Aspergillus*-specific IgG antibody level at any time neither at baseline nor at EOT or at follow-up (Figure 4). In contrast, three HIV-negative PTB patients had a positive *Aspergillus*-specific IgG antibody level of 66.6, 103, and 124 mg/L at baseline, respectively. All three PTB patients did not develop seroconversion having elevated antibody levels during tuberculosis treatment at week 12 and 17, as well as after EOT at week 26 (41.1, 174, and 45.5 mg/L, respectively). Two HIV-negative PTB patients with negative *Aspergillus*-specific IgG antibody levels at baseline of 3.08 and 11.2 mg/L became seropositive by week 26 (158 and 41.9 mg/L, respectively). One HIV-negative PTB patient had a positive *Aspergillus*-specific IgG antibody level at week 12 and 17 with 46.2 and 71.9 mg/L but became seronegative after EOT at week 26 without fungal treatment. All HIV-positive PTB patients showed very stable *Aspergillus*-specific IgG antibody levels throughout tuberculosis treatment.

In 20 PTB patients, an X-ray was available at baseline as well as at EOT. Residual cavitary lesions were documented in 45% (9/20) of PTB patients, of which five patients were HIV-positive and four patients were HIV-negative. X-rays at EOT were available in two out of five PTB patients who had a positive *Aspergillus*-specific IgG antibody level at EOT. Probable CPA was classified in one HIV-negative patient (5%; 1/20) showing a thick-walled cavity with pleural thickening (Figure 5). This patient had also the highest *Aspergillus*-IgG antibody titer with 174 mg/L from all serologically positive PTB patients at EOT.

## 4. Discussion

Our study demonstrated that the seroprevalence of *Aspergillus*-specific IgG antibodies considerably differed between HIV-positive and -negative PTB patients in Mozambique. Probable CPA was found in 5% (1/20) of PTB patients at EOT based on the *Aspergillus*-specific IgG antibody level and the radiological pattern. 

HIV-positive PTB patients had surprisingly stable *Aspergillus*-specific IgG antibody levels in the lower normal range (IQR 2.05–20.6 mg/L), while HIV-negative patients showed significantly higher values, a larger IQR, and dynamic levels of *Aspergillus*-specific IgG antibody before, during, and after the end of TB treatment (Figure 4). This difference related to the HIV status could be plausibly explained by the impaired immune system in HIV-positive individuals diminishing appropriate antibody production. Almost half of our HIV-positive PTB patients (42.5%; 17/40) had a baseline CD4 T cell count < 200 cells/μL, indicating a severe immunocompromised status. None of these patients had a positive *Aspergillus*-specific IgG antibody level, neither at baseline nor at EOT, nor follow-up. A previous study by Kwizera et al. reported comparable low baseline median *Aspergillus*-specific IgG antibody levels of 4.43 mg/L in HIV-positive PTB patients in Uganda, using the same assay as in our study [16]. However, 3 out of 76 Ugandan patients had already a positive test result > 40 mg/L at baseline. All 3 became seronegative by week 24 as an indication that the patients did not develop CPA during active TB infection. Interestingly, patients with a lower CD4 T-cell count < 100 cells/μL had significantly higher median *Aspergillus*-specific IgG antibody levels at baseline, compared to patients with a CD4 T-cell count > 100 cells/μL. This is not in accordance with our hypothesis that severely immunocompromised HIV-positive patients may have a diminished antibody response to infections with *Aspergillus* as demonstrated by our study results. None of our HIV-positive patients with a cavitary PTB had an X-ray at EOT that would have been compatible with a very high suspicion of CPA such as a fungal ball within a cavity, but 3 out of 10 HIV-positive PTB patients had an X-ray at EOT that would theoretically fulfill the radiological criteria for CPA including a thick-walled cavity, a large nodule or a cavity with pleural thickening.

Additionally, Kwizera et al. reported a significant difference between the *Aspergillus*-specific IgG levels at baseline and at week 24. Increasing *Aspergillus*-specific IgG antibody levels between the start and the end of TB treatment could be explained by an improved immune response under antiretroviral treatment. However, we did not observe this effect. although our PTB patients tested positive for HIV received comprehensive antiretroviral treatment (ART) according to National guidelines. Unfortunately, we did not have information on the CD4 T-cell count at the end of TB treatment in our HIV-positive PTB patients, which should have been higher, compared to baseline, because of ART. Further studies can benefit from determining the CD4 T-cell count at certain time points including baseline, EOT, and follow-up to allow accurate interpretation of antibody levels.

Another reason for alterations in the seroprevalence between studies could be a difference in the radiological pattern post-PTB, putting patients with residual cavities at higher risk for *Aspergillus* colonization or for chronic *Aspergillus* infection. Only 5% of the Ugandan patients in the study by Kwizera et al. had residual pulmonary cavities on the chest X-ray at week 24, while in our study, 43% (9/21) of Mozambican PTB patients had residual cavities after TB treatment [16]. Comparable high rates were also reported from Indonesia, where 61% of PTB patients had residual cavities after TB treatment [17]. Residual cavities after successful TB treatment are the most critical risk factor for developing CPA. In this data set, we have not seen any correlation concerning the frequency of cavities based on the HIV status of the PTB patients. The cumulative risk for the development of CPA following TB infection is 5–35%; however, the evidence is limited [3,16,17,18,19]. One of the most recognized studies addressing this question originates from Uganda, reporting CPA as a sequel to PTB in 4.9% (14/285) of resurveyed patients two years after TB treatment [20]. In our small pilot study, the rate of probable CPA at the end of TB treatment was 5% (1/20).

However, some limitations have to be considered. First, we only conducted X-rays, while CT cans are more sensitive for radiological evaluation of CPA. Additionally, we only conducted X-rays at EOT for two out of five patients who had positive *Aspergillus*-specific IgG antibodies at EOT. Second, we did not have precise information’s on clinical symptoms after the end of TB treatment, allowing only the classification of a probable CPA. Third, the number of patients in this study is too small to draw precise conclusions. Larger TB sequel studies are needed to address the risk of CPA as a sequel to TB. Fourth, it is very likely that there is no exact *Aspergillus*-specific IgG antibody cutoff, and therefore, the antibody values have always to be interpreted in context to symptoms, radiological pattern, and exclusion of alternative diseases. 

The combination of a chest X-ray with cavitation and serology and symptoms has a positive predictive value for CPA of 92% in patients with a history of PTB [20]. These criteria for CPA diagnosis are also found in the commonly accepted case definition for CPA for resource-constrained settings, which were established by a CPA expert panel initiated by the Global Action Fund for Fungal Infections (GAFFI) in 2016 [15]. Our study supports the idea that the relevant investigations needed for a CPA diagnosis are available or can be implemented in high PTB prevalence settings and, therefore, the combination of (a) an chest X-ray (or CT scan, if available) showing progressive cavitary infiltrates and/or a fungal ball and/or pericavitary fibrosis or infiltrates or pleural thickening, (b) chronic symptoms >3 months including weight loss, persistent cough, and/or hemoptysis, and (c) a positive *Aspergillus* IgG assay result or other evidence of *Aspergillus* infection can be used to establish CPA diagnosis also in resource-constrained settings. 

Although definite conclusions cannot be drawn, we believe that the seroprevalence and dynamic of *Aspergillus*-specific IgG antibodies may differ among Mozambican tuberculosis patients depending on their HIV status. Furthermore, probable CPA was observed in one patient at the end of TB treatment, although the number of included patients was small due to the pilot character of the study. Nevertheless, considering the global burden of TB with an estimated 10.0 million infected individuals in 2019, even lower estimates of the cumulative risk to develop CPA post TB with 5% would result in a considerable high number of patients felling ill and dying due to CPA globally. Some lesions can be learned from this pilot study. First, future studies addressing post-tuberculosis lung diseases should urgently consider CPA as a life-threatening sequel to PTB. Second, the CPA case definition for resource-constrained settings should be based on the GAFFI consensus statement including X-ray, *Aspergillus*-specific IgG antibody, and the presence of chronic symptoms. Third, it is crucial to assess both the HIV status and the CD4 T-cell count at any time point where the presence of CPA is evaluated.

## Figures and Tables

**Figure 1 jof-07-00595-f001:**
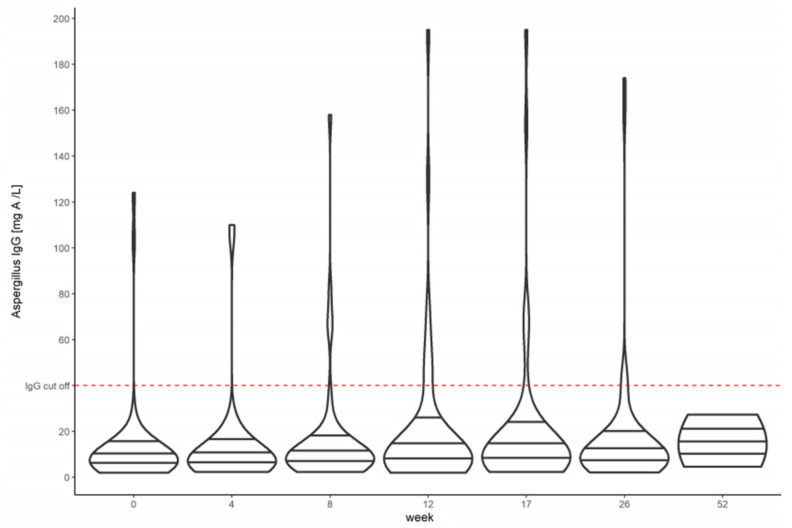
Violin plots showing the *Aspergillus*-specific IgG antibody levels before, during, and after tuberculosis treatment in HIV-positive and HIV-negative Mozambican patients with pulmonary tuberculosis.

**Figure 2 jof-07-00595-f002:**
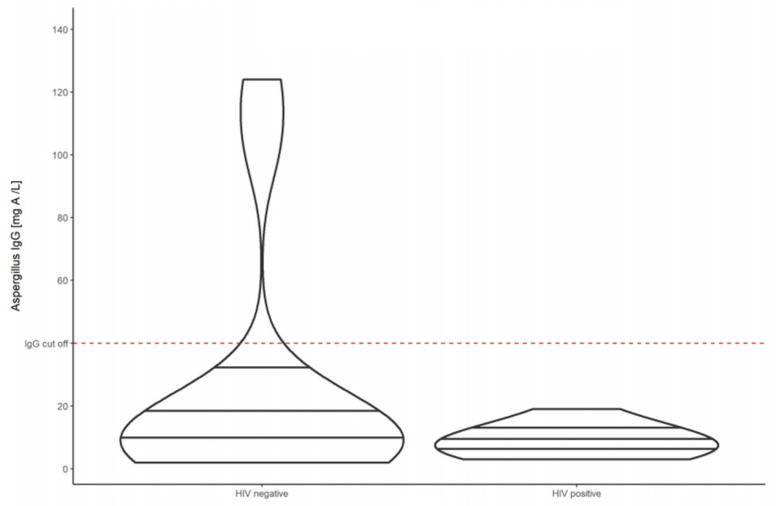
Violin plots showing the *Aspergillus*-specific IgG antibody levels of HIV-negative compared to HIV-positive patients with pulmonary tuberculosis at baseline.

**Figure 3 jof-07-00595-f003:**
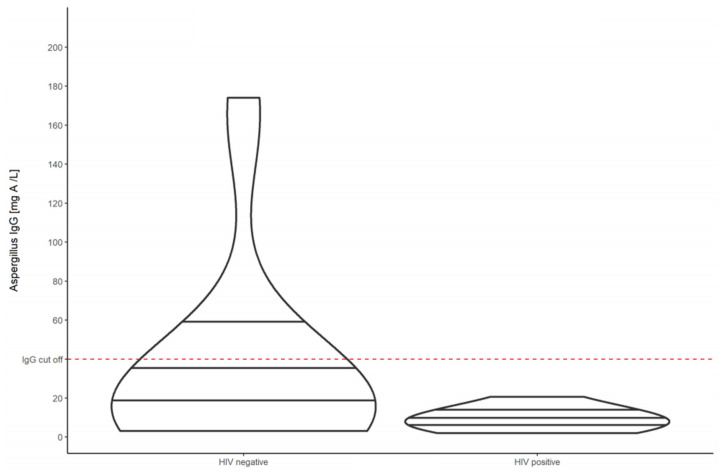
Violin plots showing the *Aspergillus*-specific IgG antibody levels of HIV-negative compared to HIV-positive patients with pulmonary tuberculosis at week 26.

**Figure 4 jof-07-00595-f004:**
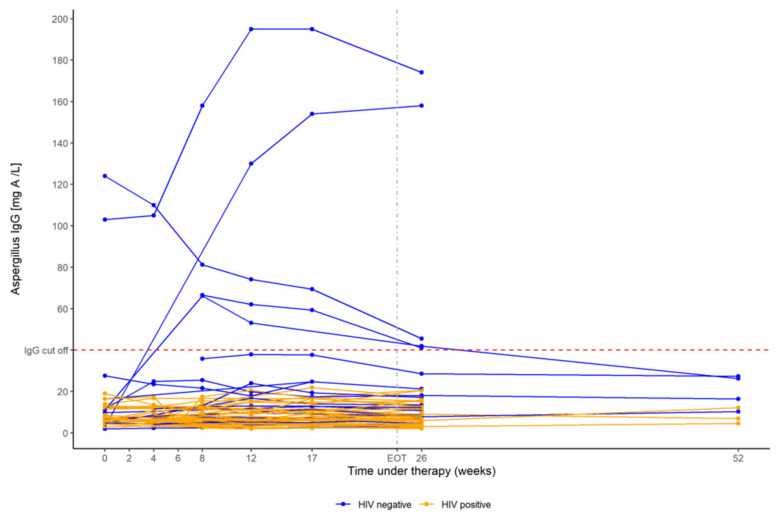
Dynamics of *Aspergillus*-specific IgG antibody levels before, during, and after tuberculosis treatment in HIV-negative (blue) and HIV-positive (orange) patients with pulmonary tuberculosis. The dashed red line marks the cutoff for a positive *Aspergillus*-specific IgG antibody level (=40 mg/L; measured on the ImmunoCAP^®^).

**Figure 5 jof-07-00595-f005:**
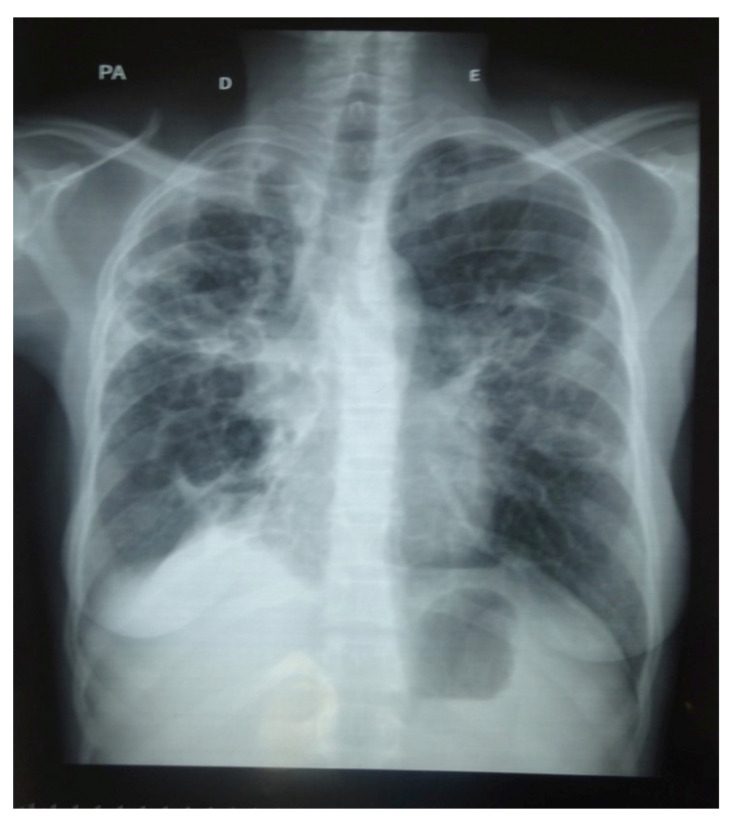
X-ray of a PTB patient showing a thick-walled cavitary lesion with pleural thickening in the right upper lung lobe at EOT having an elevated *Aspergillus*-IgG antibody titer of 174 mg/L.

**Table 1 jof-07-00595-t001:** Baseline characteristics.

Characteristics	TB Patients in Final Analysis (*n* = 61)
**Sex**	
Female, % (*n/N*)	36.1 (22/61)
**Age**	
Median (IQR)	30.0 (18, 56)
<40 years old, % (*n/N*)	75.4 (46/61)
**Body mass index**	
Median (IQR)	18.8 (13.3, 26.6)
<18.5, % (*n/N*)	45.9 (28/61)
**HIV-status**	
Positive, % (*n/N*)	65.6 (40/61)
CD4-cell count < 200/μL, % (*n/N*)	42.5 (17/40 *)
CD4-cell count 200–500/μL, % (*n/N*)	40.0 (16/40 *)
CD4-cell count ≥ 500/μL, % (*n/N*)	6.6 (4/40 *)
**History of tuberculosis**	
Yes, % (*n/N*)	6.6 (4/61)
**Rifampicin resistance**	
Present, % (*n/N*)	6.6 (4/61)
Unknown, % (*n/N*)	19.7 (12/61)
**Culture conversion**	
Until week 8, % (*n/N*)	54.1 (33/61)
Until week 26, % (*n/N*)	91.7 (55/60 ^#^)
**Haemoglobin**	
Median, g/dL (IQR)	10.9 (5.1, 15.7)
**C-reactive protein**	
Median, mg/dL (IQR)	77.3 (2.2, 322.3)

IQR, interquartile range; TB, tuberculosis, * for the 21 HIV-negative patients no CD4-cell count was documented, ^#^ data were not available for one patient.

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
