# Peer review of "Seroprevalence of Aspergillus-Specific IgG Antibody among Mozambican Tuberculosis Patients"

_jof, 2021, doi:10.3390/jof7080595_

Round 1

Reviewer 1 Report

In general, the manuscript is well written. Most data is included for the limited setting. Published data is very limited which is why this data is valuable. However, numbers are way too small to draw conclusions. Therefore, I believe this data should be added to the large-scale sequel study, or conclusions rewritten to represent an opinion in a pilot where no conclusions can be drawn.

Were these patients able to make IgG at all for other diseases, is this known? Especially the HIV positive cases with lower CD4. Continuing on this, is the figure 5 measured HIV positive IgG measurement error/baseline noise or real IgG? Do you have a healthy control cohort from the area? What does figure 5 add for your conclusions? Perhaps better to combine figure 5 in top right corner of fig 4, or remove.

2.3 - Only x-rays were used, no CT. How sensitive is the x-ray for mild cavitary TB and CPA in comparison to CT? Please add to limitations. The discussion on comparing Kwizera et al. to your study and the numbers presented must be flawed by x-ray sensitivity and observer bias. I agree the use of CT is limited in these settings, but limitations should be mentioned.

Therefore: please add a paragraph on limitations of x-ray and aspergillus IgG - you can compare to first world studies demonstrating how difficult interpretation can be comparing to symptoms. What do values mean around the cutoff? It is important to put the data in perspective, i.e. why IgG can only be interpreted in combination with symptoms and imaging, interpretating negative values. How does sensitivity differ between groups, what about other immune incompetent groups?

225 "The combination of a chest X-ray with cavitation plus serology and symptoms has a 225 positive predictive value for CPA of 92%." misses reference. In what population?

Based on these low numbers, your conclusions should be rewritten. Definite conclusions cannot be drawn from this data. "Although definite conclusions cannot be drawn, the authors believe that.." or something comparable. 

Author Response

Point 1: In general, the manuscript is well written. Most data is included for the limited setting. Published data is very limited which is why this data is valuable. However, numbers are way too small to draw conclusions. Therefore, I believe this data should be added to the large-scale sequel study, or conclusions rewritten to represent an opinion in a pilot where no conclusions can be drawn.

Response 1: We agree with the reviewer that the numbers of this pilot study are too small to draw precise conclusions. Therefore we have rewritten the conclusion sentence as well as the conclusion in the abstract. Furthermore, we have added some limitations as suggested by the reviewer.

Page 8, last paragraph: "Although definite conclusions cannot be drawn, we believe that the seroprevalence and dynamic of Aspergillus-specific IgG antibodies may differ among Mozambican tuberculosis patients depending on their HIV status."

We also agree with the reviewer that a larger study is needed and we are working on that. However, we feel that the findings of this pilot study were very important for us to improve future studies. We also included some additional limitations as suggested by the reviewer in the paragraph on limitations in the middle of page 8:

"However, some limitations have to be considered. First, we did only have X-rays, while CT cans are more sensitive for radiological evaluation of CPA. Additionally, we did only have X-rays at EOT for two out of five patients who had positive Aspergillus-specific IgG antibodies at EOT. Second, we did not have precise information’s on clinical symptoms after the end of TB treatment allowing only the classification of a probable CPA. Third, the number of patients in this study is too small to draw precise conclusions. Larger TB sequel studies are needed addressing the risk of CPA as a sequel to TB. Fourth, it is very likely that there is no exact Aspergillus-specific IgG antibody cutoff and therefore, antibody values have always to be interpreted in context to symptoms, radiological pattern and exclusion of alternative diseases."  

We also pointed out "lesions learned from this study" in the last paragraph of the discussion, which can help colleagues in the design of future CPA/TB sequel studies that are urgently needed. 

Point 2: Were these patients able to make IgG at all for other diseases, is this known? Especially the HIV positive cases with lower CD4. Continuing on this, is the figure 5 measured HIV positive IgG measurement error/baseline noise or real IgG? Do you have a healthy control cohort from the area? What does figure 5 add for your conclusions? Perhaps better to combine figure 5 in top right corner of fig 4, or remove.

Response 2: We thank the reviewer for this feedback. We can imagine that this figure may raise questions. Actually the figure shows the increase/decrease of the real Asp. IgG values for each patient, but we agree that it makes sense to remove figure 5 to enable a better readability of the manuscript. Therefore we removed figure 5.

Point 3: 2.3 - Only x-rays were used, no CT. How sensitive is the x-ray for mild cavitary TB and CPA in comparison to CT? Please add to limitations. The discussion on comparing Kwizera et al. to your study and the numbers presented must be flawed by x-ray sensitivity and observer bias. I agree the use of CT is limited in these settings, but limitations should be mentioned.

Response 3: We added this limitation. Please also see response to point 1.

Point 4: Therefore: please add a paragraph on limitations of x-ray and aspergillus IgG - you can compare to first world studies demonstrating how difficult interpretation can be comparing to symptoms. What do values mean around the cutoff? It is important to put the data in perspective, i.e. why IgG can only be interpreted in combination with symptoms and imaging, interpretating negative values. How does sensitivity differ between groups, what about other immune incompetent groups?

Response 4: 

Point 5: "The combination of a chest X-ray with cavitation plus serology and symptoms has a positive predictive value for CPA of 92%." misses reference. In what population?

Response 4: We thank the reviewer for this note. We have added the reference and also the population. "The combination of a chest X-ray with cavitation plus serology and symptoms has a positive predictive value for CPA of 92% in patients with a history of PTB [20]." 

Point 5: Based on these low numbers, your conclusions should be rewritten. Definite conclusions cannot be drawn from this data. "Although definite conclusions cannot be drawn, the authors believe that.." or something comparable. 

Response 5: We fully agree with the reviewer. We have rewritten the conclusions as suggested. Please also see response 1.

Reviewer 2 Report

The authors describe their pilot study on the utility of Aspergillus Specific IgG for the detection of CPA in patients with PTB, comparing those with and without HIV.  

In general, the manuscript was well written and reads well.  The authors understand the number of limitations of their study and really underscores the problem of lack of diagnostics in this part of the world.

Just a few comments:

Can the authors re-write the statement on lines 70 - 71 which currently reads as "Furthermore the frequency 70
of probable CPA as a sequel to PTB after EOT should be assed." - I am not entirely sure what this means.

I think the methodology can be improved.  Were any of the patients on antifungal therapy during any stage of the study?  Was there an attempt to culture Aspergillus?  The authors should state at the outset that not all patients had CXRs, CD4 counts during the study.  

The authors stated that 21 patients did not have CD4 counts done.  Were these the patients who did not have HIV?  Were CD4 counts only done on the HIV positive cohorts?

The authors mentioned that CPA can occur after end of treatment for TB.  Is there a plan to follow this cohort longitudinally to determine how many of them develop CPA 2 years, 5 years etc?  If so I would include this in the discussion/conclusion.

Author Response

Point 1: Can the authors re-write the statement on lines 70 - 71 which currently reads as "Furthermore the frequency of probable CPA as a sequel to PTB after EOT should be assed." - I am not entirely sure what this means.

Response 1: We thank the reviewer for this feedback. We absolutely agree. The sentence could be misunderstood. Therefore we simply removed this sentence, because the primary aim of this study is already mentioned in the sentence before.

Point 2: Were any of the patients on antifungal therapy during any stage of the study?  

Response 2: No patients had an antifungal therapy during the study. We have included this information.

Point 3: Was there an attempt to culture Aspergillus?

Response 3: Aspergillus cultures were not done. In resource-limited settings the performance of fungal cultures are often difficult to establish. Aspergillus is not very often detectable on culture from sputum samples in CPA patients anyway. So we decided to use the Asp. IgG antibody assay to proof mycological evidence which is also recommended by guidelines, especially for resource-limited settings like in Maputo (e.g. Denning DW. EID 2018). 

Point 4: The authors should state at the outset that not all patients had CXRs, CD4 counts during the study.  

Response 4: We have included this information.

Point 5: The authors stated that 21 patients did not have CD4 counts done.  Were these the patients who did not have HIV?  Were CD4 counts only done on the HIV positive cohorts?

Response 5: We thank the reviewer for this note. Yes, the CD4 cell count was only documented for the HIV-positive PTB patients. We have added this information in the table 1: "....* for the 21 HIV-negative patients no CD4-cell count was documented".  

Point 6: Is there a plan to follow this cohort longitudinally to determine how many of them develop CPA 2 years, 5 years etc?  If so I would include this in the discussion/conclusion.

Response 6: For this pilot study a longer follow-up period was not scheduled, but it will be an included in the large TB sequel study. Together with the other "lesions learned" from this pilot study we are convinced that we can improve the upcoming TB sequel study.

Reviewer 3 Report

The manuscript jof-1264066 presents a study on the seroprevalence of Aspergillus-specific IgG antibody among Mozambican tuberculosis patients with and without HIV infection. CPA is a Aspergillus manifestation rarely considered in an initial diagnosis; studies like this raise the awareness to this problem, especially in countries with high incidence of pulmonary tuberculosis. The manuscript is well written and shows interesting data. Some points need to be addressed:

Page 2, line 71: Since EOT is written for the first time in the manuscript, it should be written as End of Treatment (EOT). In the abstract, abbreviations should not appear.

Page 2, line 91: Was the EOT always considered as six months after the beginning of the TB treatment?

Page 3, results: The authors refer the discussion section that X-rays at EOT were available in two out of five PTB patients who had a positive Aspergillus-specific IgG antibody level at EOT. Probable CPA was classified in one HIV-negative patient (5%; 1/20) showing a thick walled cavity with pleural thickening. I suggest that this image could be added to results section to illustrate what may be observed in patients suffering from CPA.

Page 3, results: The authors refer the discussion section that “The combination of a chest X-ray with cavitation plus serology and symptoms has a positive predictive value for CPA of 92%.” In Table 1 from results’ section, the symptoms of the studied patients are not referred; furthermore, it would be interesting to perceive if the manifested symptoms were the same in both groups (HIV positive and HIV negative patients) or if there were differences.

Figure 1, legend: It should be mentioned that it refers to data that gather HIV positive and HIV negative patients.

Page 6, line 145-147: The authors stated that “One HIV-negative PTB patient had a positive Aspergillus-specific IgG antibody level at week 12 and 17 with 46.2 and 71.9 mg/l, but became sero-negative after EOT at week 26 without fungal treatment.” Is there any explanation of this?

Page 6, line 152-153: The authors refer that in 20 PTB patients an X-ray was available at baseline as well as at EOT and that X-rays at EOT were available in two out of five PTB patients who had a positive Aspergillus-specific IgG antibody level at EOT. Were these five patients included in the group of 20 previously referred?

Given the constant comparison between the group of HIV+ and HIV- patients, and to determine whether those results are, in fact, is statistically significant, statistical analysis of the results should be performed.

Page 7, line 189: Please substitute “…to infections with aspergillosis as demonstrated…” by “…to infections with Aspergillus as demonstrated…”

Page 9, line 248-250: As main conclusions, the authors refer that “it is not only crucial assessing the HIV status, but also to assess the CD4 T cell count at any time point where the presence of CPA is evaluated. The data presented by the authors point in this way since no HIV+ patients developed CPA. The authors hypothesis is that severely immunocompromised HIV+ patients may have a diminished antibody response to infections with Aspergillus.

However, in other studies, as also mentioned by the authors, the CD4 T cell count <100 cells/μl had significantly higher median Aspergillus-specific IgG antibody levels at baseline compared to patients with a CD4 T-cell count >100 cells/μl.

Thus, it is not understandable if it really necessary to assess the CD4 T cell count in HIV+ patients to measure the risk of developing CPA.

Author Response

Point 1: Page 2, line 71: Since EOT is written for the first time in the manuscript, it should be written as End of Treatment (EOT). In the abstract, abbreviations should not appear.

Response 1: We thank the reviewer for this information. We have changed it accordingly.

Point 2: Page 2, line 91: Was the EOT always considered as six months after the beginning of the TB treatment?

Response 2: Yes, the EOT was always after 6 month of treatment in this study, which was not only because of the study, but much more because of the national guidelines and real life TB management in Maputo.

Point 3: I suggest that this image could be added to results section to illustrate what may be observed in patients suffering from CPA.

Response 3: We have included the X-ray as suggested by the reviewer including a figure legend.

Point 4: Page 3, results: The authors refer the discussion section that “The combination of a chest X-ray with cavitation plus serology and symptoms has a positive predictive value for CPA of 92%.” In Table 1 from results’ section, the symptoms of the studied patients are not referred; furthermore, it would be interesting to perceive if the manifested symptoms were the same in both groups (HIV positive and HIV negative patients) or if there were differences.

Response 4: That is true. Unfortunately, the symptoms of the PTB patients were not documented at EOT. We have stated this as a limitation of our study. That is also a reason why we only described the performance of the Aspergillus IgG antibodies, rather than drawing exact conclusions on the rate of CPA. However, this is one of the "lesions learned" from this pilot study and which will be improved in the large TB sequel study.

Point 5: Figure 1, legend: It should be mentioned that it refers to data that gather HIV positive and HIV negative patients.

Response 5: We have added this information in the figure legend 1.

Point 6: Page 6, line 145-147: The authors stated that “One HIV-negative PTB patient had a positive Aspergillus-specific IgG antibody level at week 12 and 17 with 46.2 and 71.9 mg/l, but became sero-negative after EOT at week 26 without fungal treatment.” Is there any explanation of this?

Response 6: Unfortunately not. This observation were also reported by other colleagues, but it is not really understood. However, it shows that the Asp. IgG antibody value showed always be interpreted in context to the radiological pattern, symptoms and exclusion of alternative diagnosis. It is definitely not a precise biomarker for CPA diagnosis, but an important piece of the puzzle to indirectly proof mycological evidence, especially in resource-limited settings, where classical fungal culture technics are often difficult to establish.     

Point 7: Given the constant comparison between the group of HIV+ and HIV- patients, and to determine whether those results are, in fact, is statistically significant, statistical analysis of the results should be performed.

Response 7: We also had this idea, however, our statistician recommended not to do it, because the numbers are too small to draw exact conclusions. Therefore he recommended descriptive analyses including median, IQR, percentages, violin blots, etc. as we did in this version. If that is okay we would be pleased to keep it as it is, however if the reviewer prefers to include it than we of course will do that.

Point 8: Page 7, line 189: Please substitute “…to infections with aspergillosis as demonstrated…” by “…to infections with Aspergillus as demonstrated…”

Response 8: We thank the reviewer for the note. We have changed this accordingly.

Point 9: However, in other studies, as also mentioned by the authors, the CD4 T cell count <100 cells/μl had significantly higher median Aspergillus-specific IgG antibody levels at baseline compared to patients with a CD4 T-cell count >100 cells/μl. Thus, it is not understandable if it really necessary to assess the CD4 T cell count in HIV+ patients to measure the risk of developing CPA.

Response 9: We absolutely agree with the reviewer. The limited evidence published so far is not conclusive. We also believe that this aspect has to be investigated in a larger study to get more robust data. However, we also believe that our observation of HIV-positive patients with low CD4 cells may not able to effectively produce Asp-IgG antibodies and that this could be plausible. Nevertheless, we absolutely accept that the number of patients included in our pilot study is too small to draw exact conclusions. Therefore, we hope that future studies can help to improve the understanding on that aspect.